# CRISPR/Cas9 and active genetics-based trans-species replacement of the endogenous *Drosophila kni*-L2 CRM reveals unexpected complexity

Xiang-Ru Shannon Xu[1†], Valentino Matteo Gantz[1†], Natalia Siomava[2‡], Ethan Bier[1]*

[1]Section of Cell and Developmental Biology, University of California San Diego, La Jolla, California; [2]Department of Developmental Biology, Johann-Friedrich-Blumenbach Institute for Zoology and Anthropology, Georg-August-University of Göttingen, Ernst-Caspari-Haus (GZMB), Göttingen, Germany

**Abstract** The *knirps* (*kni*) locus encodes transcription factors required for induction of the L2 wing vein in *Drosophila*. Here, we employ diverse CRISPR/Cas9 genome editing tools to generate a series of targeted lesions within the endogenous cis-regulatory module (CRM) required for *kni* expression in the L2 vein primordium. Phenotypic analysis of these '*in locus*' mutations based on both expression of Kni protein and adult wing phenotypes, reveals novel unexpected features of L2-CRM function including evidence for a chromosome pairing-dependent process that promotes transcription. We also demonstrate that self-propagating active genetic elements (CopyCat elements) can efficiently delete and replace the L2-CRM with orthologous sequences from other divergent fly species. Wing vein phenotypes resulting from these trans-species enhancer replacements parallel features of the respective donor fly species. This highly sensitive phenotypic readout of enhancer function in a native genomic context reveals novel features of CRM function undetected by traditional reporter gene analysis.
DOI: https://doi.org/10.7554/eLife.30281.001

*For correspondence:
ebier@ucsd.edu

[†]These authors contributed equally to this work

Present address: [‡]College of Medicine, Howard University, Washington, D.C., United States

## Introduction

The *Drosophila* wing provides an excellent model system for studying the relationship between gene expression and morphogenesis since it is a well-studied, two-dimensional, developmental system that generates invariant morphological features, such as veins and sensory organs (*Sturtevant et al., 1993*). Wing veins provide structural support necessary for flight and supply nutrients to sensory organs that coordinate wing beat motions (reviewed in [*Bier, 2000*]). The number and positioning of veins are highly selected characteristics that vary among dipterans, holding evolutionary significance pertinent to the divergence of insect species and their modes of flight (reviewed in [*Bier, 2000*]).

*Drosophila* wing venation is determined by an intricate gene regulatory network (GRN) acting during larval and early pupal stages to position the five major longitudinal veins (L1-L5) and to sculpt the overall final shape of the wing. Secreted morphogens, including Hedgehog (Hh) and Decapentaplegic (Dpp), activate the expression of primary response genes, such as *spalt* (*sal* = *salm* + *salr*) and *optomotor-blind* (*omb*, also known as *bifid*), in broad domains. These patterning genes have been proposed to subsequently induce the formation of vein primordia along their borders by 'for-export-only-signaling' mechanisms (reviewed in [*Bier, 2000*]) and cross-repressive interactions (*Al Khatib et al., 2017*; *Martín et al., 2017*).

**eLife digest** Gregor Mendel was an Austrian monk and botanist whose work with pea plants in the 19th century founded the field of genetics. Though he was not aware of genes at the time, Mendel essentially worked out that pea plants had two copies of each gene, and that each copy had a 50% chance of being passed on to any one offspring. Yet not all genes actually follow this pattern of inheritance.

In 2015, researchers reported that they had used components of the CRISPR/Cas9 genome editing system to edit genes so that they could propagate in a "Super-Mendelian" fashion. Indeed, when it was engineered into fruit flies, any parent carrying this active genetic element passed it on to almost every offspring. Active genetic elements have potential applications in many different fields of scientific research. These include providing new ways to explore how genes control the formation and activity of different organisms.

Now, Xu, Gantz et al. – including the two researchers involved in the 2015 work – have used a new active genetic element called a CopyCat element and more traditional genome editing to analyze the control of a gene that coordinates the formation of a simple structure in a fruit fly – a vein in the wing. The goal was to understand which sections of DNA controlled where and when genes are activated to result in this structure being reliably located in its correct position.

First, Xu, Gantz et al. used genome editing to make mutations in a stretch of DNA that regulates the gene involved in wing vein formation. The effects of these mutations unexpectedly suggested that pairs of chromosomes might be interacting to control the activity of this gene. This was something that had not been seen before, which shows the advantage of editing a gene's regulatory sequence at its normal location within the genome.

Next, Xu, Gantz et al. used the CopyCat tool to delete the regulatory sequence and replace it with sequences from three other species of flies. When the sequence was replaced with that of a housefly, a complete vein formed but it was further forward than normal for a fruit fly, and more closely matched the position of the wing vein in a housefly. These findings show how gene activity can affect the position of a simple structure; they also suggest that this strategy could help scientists to understand how the genomes of different species have evolved.

Xu, Gantz et al. hope these advances will encourage other researchers to use active genetic elements in a broad range of organisms to enable and accelerate their research. Since these tools fundamentally change the rules of genetic inheritance, they have many applications beyond research too. These applications are not without their risks and would need careful consideration, but could include engineering wild mosquito populations to combat diseases like malaria, dengue fever, chikungunya and Zika.

DOI: https://doi.org/10.7554/eLife.30281.002

The L2 vein primordium forms along the anterior border of the *sal* domain (*Sturtevant et al., 1997*), where the zinc finger transcription factors, *knirps* and its neighboring homolog, *knirps-related* (*knrl*) are induced and direct the L2 vein development program (*Lunde et al., 1998*). Genomic lesions associated with several independently generated *radius incompletus* (*ri*) regulatory alleles of the *kni* locus, include deletions or point mutations within a 4.8 kb fragment upstream of the *kni* coding region that greatly reduce or eliminate expression of reporter genes in the L2 primordium and result in L2 vein truncation phenotypes (*Lunde et al., 2003*). Thus, the *kni*[ri[1]] mutant (*Figure 1D*) contains a 252 bp deletion, while the *kni*[ri[53j]] allele (*Figure 1E*) consists of a single point mutation (C to A within the same region deleted in *kni*[ri[1]] mutants), indicating the importance of a single nucleotide residue in this position. A minimal ~1.4 kb *kni* CRM fragment (EX - *Figure 1A*) was defined by deletion analysis, driving reporter gene expression in a pattern similar to that of the full 4.8 kb fragment. However, further truncation of that region to a 0.69 kb fragment (EC - *Figure 1A*) resulted in strong ectopic expression in anterior and posterior domains of the imaginal disc, suggesting that the reciprocal fragment contained negative regulatory elements responsible for repressing *kni* expression in peripheral regions of the wing (*Lunde et al., 2003*).

The prior experiments with CRM-reporter gene fusions summarized above, provided important information regarding the nature and organization of CRM sub-modules, but did not provide a clear

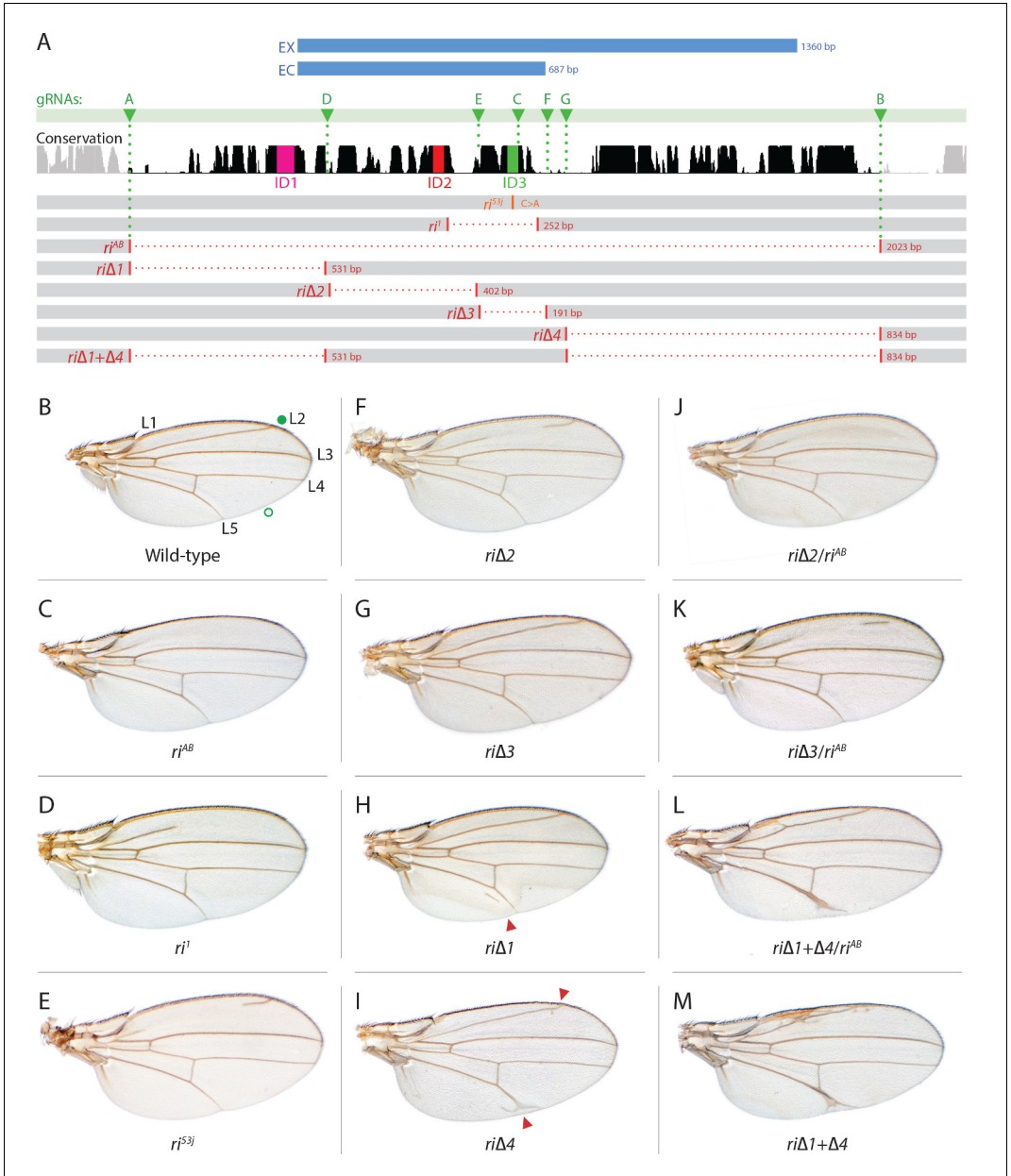

**Figure 1.** Fine scale *in-locus* deletion analysis of the *D mel. kni* L2-CRM. (**A**) Map of *kni* L2-CRM and mutants. Diagram displays features of the cis-regulatory module (CRM) of the *kni* locus that restricts expression of the closely related *kni* and *knrl* genes in the L2 vein primordium. The black bar-scape indicates regions of sequence conservation, the green bar, arrows and dotted lines indicate the positions of guide RNAs (gRNAs, lettered A-G) used for dissection of this locus, the blue bars at the top indicate CRM-reporter gene fusions that have been previously analyzed (*Lunde et al., 2003*). The region present in EX and absent in EC were shown to contain sequences mediating repression of kni expression in peripheral regions of the wing. The '*in-locus*' deletions generated in this study as well as known lesion (*kni*$^{ri53}$ and *kni*$^{ri1}$) are represented as gaps with a dotted red line along with details of the exact size of the deletion. These mutants were generated by injecting embryos expressing Cas9 with plasmids encoding the indicated gRNAs. (**B–M**) Adult wings of the following genotypes: (**B**) wild-type (WT) labeled with a filled or open green circle indicating anterior and posterior Sal borders, respectively, where Kni is expressed (see *Figure 2A,M–O*); C) homozygous AB deletion = *kni*$^{riAB}$/*kni*$^{riAB}$. (**D**) *kni*$^{ri1}$/*kni*$^{ri1}$; (**E**) *kni*$^{ri53j}$/*kni*$^{ri53j}$; (**F**) *kni*$^{riΔ2}$/*kni*$^{riΔ2}$; (**G**) *kni*$^{riΔ3}$/*kni*$^{riΔ3}$; (**H**) *kni*$^{riΔ1}$/*kni*$^{riΔ1}$. (**I**) *kni*$^{riΔ4}$/*kni*$^{riΔ4}$; (**J**) *kni*$^{riΔ2}$/*kni*$^{riAB}$; (**K**) *kni*$^{riΔ3}$/*kni*$^{riAB}$; (**L**) *kni*$^{riΔ1+Δ4}$/*kni*$^{riAB}$; (**M**) *kni*$^{riΔ1+Δ4}$/*kni*$^{riΔ1+Δ4}$.

DOI: https://doi.org/10.7554/eLife.30281.003

The following figure supplement is available for figure 1:

**Figure supplement 1.** Wing phenotypes associated with fortuitously recovered CRISPR L2-CRM alleles.
DOI: https://doi.org/10.7554/eLife.30281.004

link to gene function in the context of the endogenous *kni* locus. The recent advent of highly efficient CRISPR/Cas9 (clustered randomly interspaced palindromic repeats/CRISPR-associated protein 9) genome editing tools has provided a method to precisely alter CRM sequences and observe the resulting effects on development of the final L2 vein structure in the adult wing, permitting more sensitive decoding of CRM functionality. The type II CRISPR/Cas9 endonuclease system evolved in bacteria as a defensive measure against viruses to cleave invading foreign DNA (*Barrangou et al., 2007*). This natural defense system has been engineered into a bipartite genome editing tool, consisting of a guide RNA (gRNA) that directs double stranded DNA cleavage by the Cas9 endonuclease at specific locations within the genome (*Jinek et al., 2012*) and has been adapted for use in a myriad of organisms (reviewed in: (*Bassett and Liu, 2014*; *Doudna and Charpentier, 2014*; *Hsu et al., 2014*; *Overcash et al., 2015*; *Sander and Joung, 2014*; *Sternberg and Doudna, 2015*; *Zhang et al., 2014*), including *Drosophila* (*Bassett et al., 2013*; *Gratz et al., 2013a*; *Kondo and Ueda, 2013*; *Ren et al., 2013*; *Yu et al., 2013*). Following gRNA/Cas9 cleavage at a given target site, mutations can be generated at that genomic location via one of two pathways: error-prone non-homologous end joining (NHEJ) DNA repair, which typically generates small insertions and/or deletions (indels) at the cut site, or the more precise homology-directed repair (HDR) pathway that copies sequences from a homologous DNA template (*Gratz et al., 2013*). With the advent of CRISPR technology, these methods have also been employed to generate 'active genetic' elements that are copied in the germline via HDR and inherited in a Super-Mendelian fashion (reviewed in [*Gantz and Bier, 2016*]).

In the current study, we employed CRISPR/Cas9 to generate mutations in the endogenous *Drosophila kni* L2-CRM and analyzed the phenotypic effects of these lesions on adult wing vein pattern. We generated a series of targeted sequential deletions spanning a ~ 2 kb segment encompassing the previously identified minimal 1.4 kb L2-CRM fragment plus additional well-conserved adjacent sequences. We also recovered and analyzed a variety of additional non-targeted mutations associated with imprecise lesions that were most likely generated by NHEJ. These studies provide a much higher resolution view of CRM function by linking CRM lesions in their native chromosomal context to precise phenotypic outputs. In addition, we validate the high efficiency of homology-directed, CRISPR-mediated, site-specific transgenesis as a viable alternative to traditional transgenesis methods (e.g. P-element or ΦC31-based systems). We also introduce the use of active genetic elements we refer to as CopyCat cloning vectors that can be inserted at determined locations and are copied with high efficiency from one parental chromosome to another during germline transmission, in the presence of an unlinked source of Cas9. Finally, we show that such active CopyCat vectors can be used for efficiently replacing the L2-CRM with homologous sequences from other fly species, resulting in alternative patterns of L2 vein placement. These new active genetics tools should greatly accelerate detailed combinatorial analysis of gene regulatory networks in a variety of experimental settings, as well as analyzing the function of exogenous DNA sequences found among diverged species.

## Results

### *In locus* deletion of the L2-CRM eliminates the adult L2-vein and Knirps expression in the larval L2-primordium

As summarized in the introductory section, well-validated CRISPR/Cas9 genome editing tools now make it routine to generate genome alterations via random mutagenesis near guide RNA (gRNA) cleavage sites (via the NHEJ pathway) or to create targeted edits based on templates carrying homologous sequences on either side of the gRNA-directed cut site (via homology direct repair - HDR). We employed the latter precise approach to generate a series of deletions of the endogenous L2-CRM. We utilized two gRNAs directing Cas9 cleavage on each side of a specific region to be deleted, as well as single-stranded oligonucleotides donors (ssODN) with 60 bp of homology to either DNA end flanking the region targeted for deletion (see Materials and Methods). We first used two gRNAs (gRNA-A and gRNA-B) to delete a 2023 bp regulatory region encompassing the previously identified 1.4 kb minimal L2-CRM. The choice of this region was based primarily on significant sequence conservation among closely related flies which extended beyond both borders of the fragment (*Figure 1A*) analyzed in our original L2-CRM reporter studies (*Lunde et al., 2003*). When

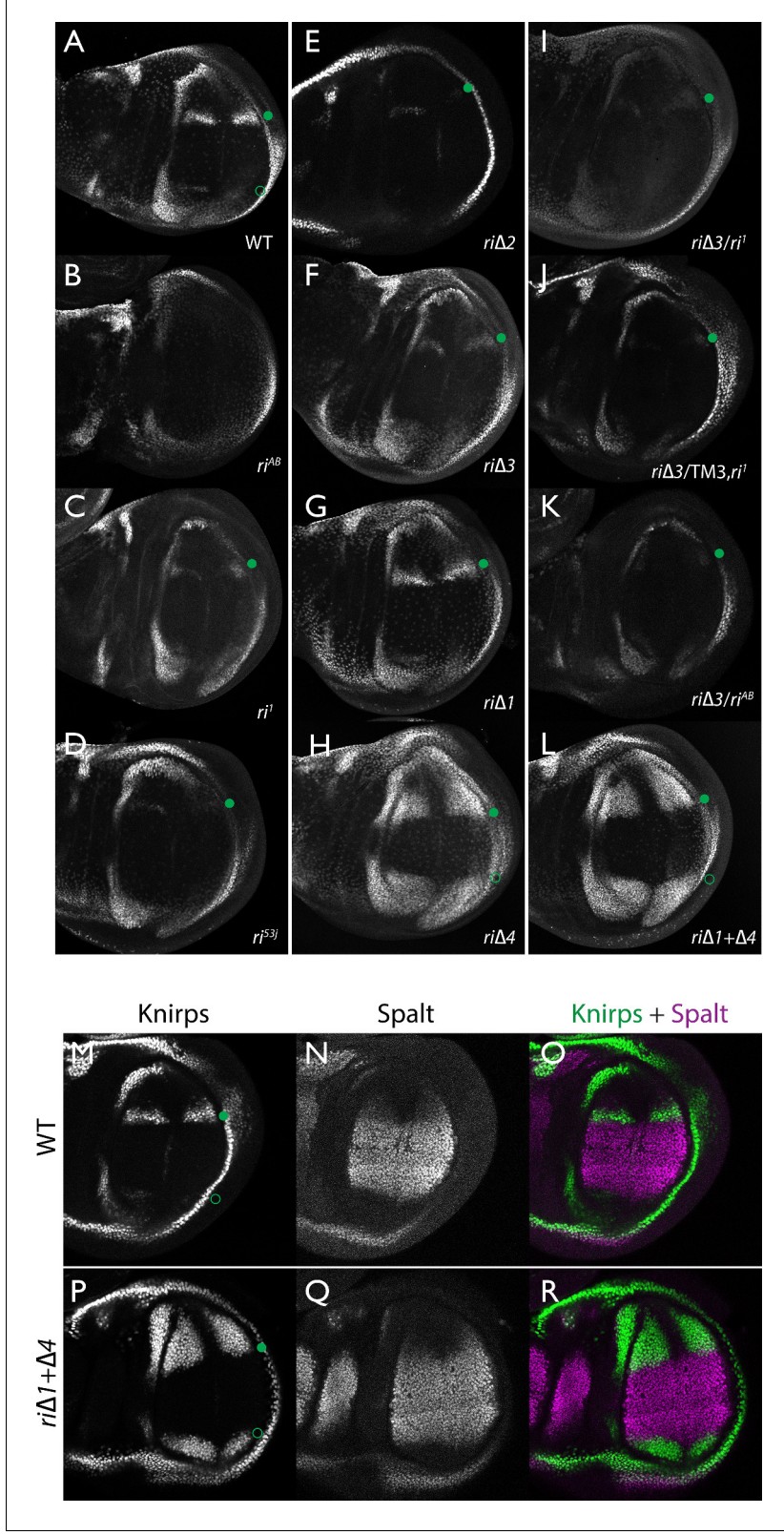

**Figure 2.** Endogenous Kni expression resulting from CRISPR generated in-locus CRM mutations. Kni expression in late third instar wing imaginal discs was determined by staining with a polyclonal anti-Kni antibody alone (**A–L**) or in combination with an anti-Spalt antibody (**M–R**). Genotypes are specified by the labeling of each panel, marked with a filled green circle indicating location of anterior Kni expression and an open green circle indicating faint

*Figure 2 continued on next page*

*Figure 2 continued*

posterior expression along the Sal border. In panels (**M–R**), stains for Kni and Spalt are shown separately as black and white images and combined as green (Kni) and magenta (Sal) images.

DOI: https://doi.org/10.7554/eLife.30281.005

The following figure supplement is available for figure 2:

**Figure supplement 1.** Knirps and Spalt expression in prepupal wings.

DOI: https://doi.org/10.7554/eLife.30281.006

rendered homozygous, the *kni^ri-AB* allele resulted in complete elimination of the L2 vein (*Figure 1C*) suggesting that it represents a true, null allele of the L2-CRM function. Consistent with this phenotypic assessment, Kni protein expression was undetectable in the L2 primordium of *ri-AB* homozygous third larval instar wing imaginal discs (*Figure 2B*) or early pre-pupal wings (*Figure 2—figure supplement 1*).

## Sequential deletions spanning the endogenous L2-CRM define additional sequences required for *Kni* activation and repression

As diagrammed in *Figure 1A*, we also generated strains of flies carrying a series of four smaller deletions (*riΔ1*-*riΔ4*) spanning the ~2 kb AB interval, each eliminating contiguous regions of sequence conservation. We also indicate three short regions that display a particularly high degree of sequence conservation across Drosophilids and other Schizophora flies, that we refer to as hyperconserved identity islands (labeled ID1, ID2 and ID3, respectively, magenta, red and green boxes in *Figure 1A* also indicated in the multispecies alignment depicted in Figure 5A, and at the DNA sequence level in Figure 5—figure supplement 1). The generated deletions were tested for phenotypes in trans-heterozygosity with the *riAB* deletion allele and in the homozygous condition. Consistent with prior studies that identified a central region required for CRM activity (i.e. a 252 bp region deleted in *ri1* (*Figure 1D*) that contains essential Scalloped activator binding sites [*Lunde et al., 2003*]), the *riΔ2* and *riΔ3* deletions, which both partially overlap with the region deleted in *ri1*, produced strong vein-loss phenotypes similar to those of *ri1* when placed in trans to *riAB* (*Figure 1J,K*). Also, consistent with the prior mapping of a repressor domain to distal sequences missing in the EC fragment (*Figure 1A*: blue box), the homozygous *riΔ4* deletion produced a mild ectopic vein phenotype in peripheral regions of the wing (*Figure 1I*). The homozygous *riΔ1* deletion produced a similar mild ectopic vein phenotype (*Figure 1H*) lending support to our suspicion mentioned above that conserved sequences that were not included in our previous reporter analysis, contribute to the overall CRM function. Since both the *riΔ1* and *riΔ4* deletions produced mild ectopic vein phenotypes, we generated a composite *riΔ1+Δ4* deletion (by generating the *riΔ1* deletion in the background of *riΔ4*) to determine whether these separated domains of the L2 CRM might act in concert. We found that this was indeed the case, as the homozygous combined *riΔ1+Δ4* deletion produced a much more pronounced phenotype in which large thickened veins formed around L2 and L5 (*Figure 1M*), a phenotype that was similar albeit less pronounced in trans-heterozygotes over the AB deletion (*riΔ1+Δ4/riAB* *Figure 2L*). Consistent with this phenotype, significant levels of ectopic Kni expression were observed in the third instar imaginal discs of *riΔ1+Δ4* homozygous mutants in both anterior and posterior domains of the wing pouch (*Figure 2L*), in a pattern similar to that driven by the EC-lacZ reporter construct analyzed previously (*Lunde et al., 2003*). The domains of ectopic Kni expression in *riΔ1+Δ4* homozygotes are largely complementary to those of the Spalt transcription factor (*Figure 2P–R*) as is the case for wild-type Kni expression (*Figure 2M–O*), indicating that binding sites for Spalt to repress Kni expression in central regions of the wing, are likely retained in the remaining CRM sequences. Interestingly, the pattern of ectopic Kni expression in homozygous *riΔ4* single mutant wing discs (*Figure 2H*) was similar in both pattern and level to that in *riΔ1+Δ4* double mutant discs (*Figure 2L*), and only mild ectopic Kni expression was observed in *riΔ1* mutant discs (*Figure 2G*). Slightly later in development during pre-pupal stages, however, narrower domains of high level Kni expression are observed in *riΔ1+Δ4* double mutant (*Figure 2—figure supplement 1R*) in comparison to the persisting broader domains of Kni expression present in the *riΔ1* single mutant (*Figure 2—figure supplement 1O*). These later differences in Kni expression may help account for differences in the severity of ectopic wing vein phenotypes in the *riΔ4* single mutant

versus the double *riΔ1+Δ4* mutant(see Discussion). Thus, a combination of the Kni expression patterns and resulting adult morphological phenotypes provide more informative distinctions between these various mutants than can be ascertained by either measure alone.

## Chromosomal pairing may play a role in L2-CRM activity

In addition to phenotypes described above that were consistent with and extended previous observations, we also encountered unexpected results when testing the effect of homozygous *riΔ2* and *riΔ3* deletions. In contrast to the strong vein-loss phenotypes associated with these deletions in trans to the complete *riAB* deletion described above (*Figure 1J,K*), the phenotypes of these lesions were greatly reduced when homozygous (*Figure 1F,G*). This disparity was particularly striking for the homozygous *riΔ3* deletion, which generated an almost wild-type phenotype, recognizable only by a subtle meandering or less pigmented L2 vein (*Figure 1G*). This very weak phenotype, which was observed in four independently derived *riΔ3* mutant strains, is particularly surprising given that a single base pair change within that region associated with the *ri53j* allele leads to a substantial vein loss phenotype when homozygous ([*Lunde et al., 2003*]; *Figure 1E*). The recovery of these two deletions that produced surprisingly mild homozygous phenotypes suggested the possibility that a pairing-dependent process might contribute to sustaining CRM function. Consistent with this hypothesis, trans-heterozygotes of the *riΔ3* deletion over the point mutant ri53j allele (*Figure 3J*), - a configuration predicted to disrupt precise alignment of homologous chromosomes - resulted in a pronounced vein-loss phenotype. We tested the pairing hypothesis more directly using the classic chromosomal inversion comparison by placing the *riΔ3* deletion in-trans to the *ri1* mutant either in the context of a native third chromosome or in a multiply inverted balancer chromosome (*TM3, ri1*) that greatly reduces pairing-dependent phenomena such as transvection (i.e. wherein the CRM on one chromosome acts to promote transcription from the homologous chromosome [*Lewis, 1954*]). Consistent with the pairing-based mechanism, the *riΔ3/ri1* phenotype (*Figure 3B*) was significantly weaker than that observed in *riΔ3/TM3, ri1* individuals (*Figure 3F*), an effect that was highly consistent (see *Figure 3—figure supplement 1* for comparative L2 vein arrays of these two genotypes). A similar, albeit less dramatic, difference was also observed between the *riΔ2/ri1* (*Figure 3A*) and *riΔ2/TM3ri1* (*Figure 3E*) phenotypes. Reflective of the adult wing phenotypes, Kni expression in larval wing was only modestly reduced in *riΔ3* homozygotes (*Figure 2F*), was further reduced in *riΔ3/ri1* discs (*Figure 2I*) and virtually undetectable in either *riΔ3/TM3ri1* (*Figure 2J*) or *riΔ3/riAB* trans-heterozygous discs (*Figure 2K*). As a final test of the CRM-pairing hypothesis we crossed the *riΔ2* and *riΔ3* mutants to each other and observed a substantial vein-loss phenotype in the *riΔ2/riΔ3* trans-heterozygotes, again consistent with a disruption of local chromosomal pairing of the remaining activating elements. Cumulatively, these data provide evidence for a pairing-dependent interaction between CRM sequences on homologous chromosomes that promotes CRM activity (see Discussion; *Figure 3M–R*)

## Novel fortuitously generated mutations with unexpected phenotypes

In addition to recovering strains of flies carrying the targeted deletions mentioned above, we also identified an ample collection of imprecise or fortuitously generated mutations that most likely resulted from DNA repair by the NHEJ or other break-repair pathways responding to single or double cleavage events at various locations (*Figure 1—figure supplement 1A*). For example, a deletion (*riΔ3.35*) that spans sequences removed in *riΔ4* and includes small indels results in a variable loss of vein phenotype when placed in trans to *riAB* (*Figure 1—figure supplement 1B,C*) but has a wild-type phenotype when homozygous (*Figure 1—figure supplement 1D*) in contrast to the extra vein phenotype manifest by *riΔ4* homozygotes (*Figure 1I*). Additionally, we recovered mutants which were generated with the use of a single gRNA, gRNA-C, targeting a conserved region (see *Figure 1A* and *Figure 1—figure supplement 1A*). They included deletions (EV 5–1 and EV 7–1) that have very similar breakpoints but result in differing degrees of vein-loss (*Figure 1—figure supplement 1E,F*), and the addition of a single nucleotide at the edge of the ID3 conserved region (EV 9–2) which has a phenotype that varies from wild type to an *ri*-like vein phenotype (*Figure 1—figure supplement 1G*). In another case (*riΔ1.1*), a 65 bp sequence that has been copied and inserted upstream causes a mild ectopic vein phenotype (*Figure 1—figure supplement 1H*). Perhaps not surprisingly, a deletion removing sequences spanning *riΔ1* to a portion of *riΔ4* (*riΔ1.6*) gives a

composite partial vein loss and ectopic vein phenotypes (*Figure 1—figure supplement 1I,J*), while a mutant with a deletion spanning *riΔ3-Δ4* plus a 25 bp insertion (*riΔ3.38*) causes an extreme vein-loss phenotype similar to that of *riAB* (*Figure 1—figure supplement 1K,L*). Although further analysis of several of these mutants will be necessary to fully understand the basis for associated venation phenotypes, we note two salient features of this mutant class. First, a variety of mutations preferentially affect portions of the L2 vein along the proximo-distal (PD) axis, revealing a previously unappreciated role of the PD patterning system in contributing to L2 morphogenesis. Second, in several cases, apparently minor variations in the positions of chromosomal breakpoints or small indels, result in markedly altered final vein patterns that may reveal important effects of specific sequences or requirements for precise spacing between functional motifs. Our recovery of several mutants with novel unanticipated phenotypes highlights the strength of an *in locus* analysis as all these mutations, even those resulting in very mild phenotypes, were readily identified phenotypically.

## Efficient Cas9-dependent germline transmission of an active *riAB* CopyCat element

We recently demonstrated an active genetic process, which we refer to as the mutagenic chain reaction (MCR), to generate both somatic mutations and meiotic gene-drive (*Gantz and Bier, 2015*). The key feature of this process is the integration of a vector containing a gRNA and a Cas9 transgene at the precise genomic location of the gRNA-directed cleavage site. We also have proposed a split-drive configuration we refer to as CopyCat elements (*Gantz and Bier, 2016*), in which the Cas9 source is provided in-trans in a standard Mendelian fashion and the CopyCat element carries one gRNA to target insertion of a gene cassette or two gRNAs to simultaneously trigger cassette insertion and deletion of the region in between the two gRNA cleavage sites. CopyCat cloning vectors could, in principle, be used as versatile elements for germline transformation, which once inserted, are passed down to the progeny in a Super-Mendelian fashion (>>50% inheritance) in the presence of a separate Cas9 source that can be later segregated away. This conditional control of Super-Mendelian inheritance can facilitate combinatorial genetic schemes by circumventing constraints imposed by Mendelian inheritance (i.e. random chromosomal segregation and linkage) (*Gantz and Bier, 2016*).

We tested the concept of a two gRNA CopyCat element by generating an active *riAB* allele. A CopyCat *riAB* vector carrying the two gRNA-expressing genes required for generating the AB deletion, an eGFP marker gene, flanked by 1 kb homology arms corresponding to the *kni* locus abutting the two gRNA cut sites was inserted into the genome by co-injection with a Cas9-producing plasmid into embryonic polar plasm (*Figure 4A*). Transformant lines carrying this and other similar CopyCat constructs described below were recovered at frequencies similar to those typical of germline transformation using either P-element vectors or the ΦC31 recombinase system, exemplifying how Cas9-mediated site-directed transgenesis can serve as a viable alternative to traditional methods of germline transformation. We combined the *riCC-AB* allele with the previously characterized *y1-MCR* element, which contained a germline-expressed Cas9 gene, a gRNA targeting the *yellow* locus, and homology arms that precisely flank the genomic cleavage site. Following the cross of *riCC-AB* males to *y1-MCR* females (*Figure 4B*), resulting female progeny were crossed to *w-* males to evaluate the transmission of the *riCC-AB* allele in their progeny, as well as the generation of somatic phenotypes in female progeny. The results of nine such crosses carried out in parallel are shown in *Figure 4C,D*, all of which demonstrated a high rate of somatic mutagenesis of the *yellow* locus, indicating the efficient activity of the *y-1MCR* element. In four crosses, the *riCC-AB* allele was copied with 100% efficiency to F2 progeny, in three crosses the efficiency averaged 89%, and in two crosses the element was inherited at Mendelian frequencies. Regardless of the rate of germline transmission noted in these various crosses, the observable L2 truncation frequencies was indicative of somatic activity of the *riCC-AB* CopyCat and Cas9 alleles which were very similar and approximated that observed in their F1 female parent (~70%). We conclude that in this system the *riCC-AB* double-cut CopyCat element typically propagates via the germline with high efficiency and that the somatic and germline activities of the *riCC-AB* element seem to be separable events. This effect is probably due to high levels of Cas9 protein produced by the *vasa* driver in the egg, that can freely diffuse prior to cellularization, causing mutations affecting large regions of the organism, consistent with observations reported by Port et al. (*Port et al., 2014*).

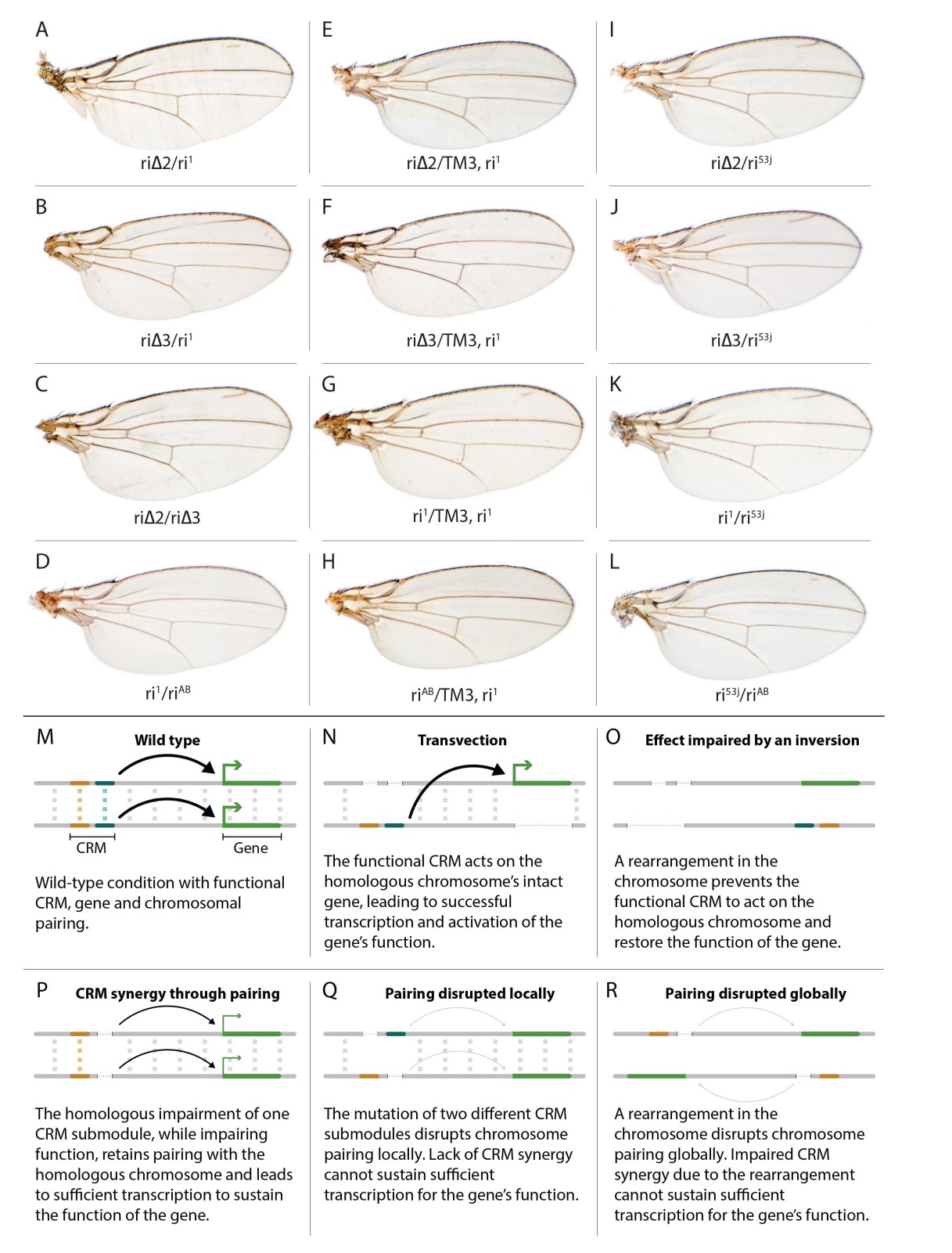

**Figure 3.** Evidence for potential chromosomal pairing dependent interactions between *kni* L2-CRM mutants. (A–L) adult wings. (A) *riΔ2/ri1* trans-heterozygote = *kni*[riΔ2]/*kni*[ri1]; (B) *kni*[riΔ3]/*kni*[ri1]; (C) *kni*[riΔ2]/*kni*[riΔ3]; (D) *kni*[ri1]/*kni*[riAB]; (E) *kni*[riΔ2]/TM3,*kni*[ri1]; (F) *kni*[riΔ3]/TM3,*kni*[ri1]; (G) *kni*[ri1]/TM3,*kni*[ri1]; (H) *kni*[riAB]/TM3,*kni*[ri1]; (I) *kni*[riΔ2]/*kni*[ri53j]; (J) *kni*[riΔ3]/*kni*[ri53j]; (K) *kni*[ri1]/*kni*[ri53j]; (L) *kni*[ri53j]/*kni*[riAB]. (M–R) Transvection vs. CRM-synergy models: (M) Wild-type interactions between two hypothetically positively acting CRM sub-modules on homologous chromosomes. Such pairing-dependent interactions are represented

*Figure 3 continued on next page*

*Figure 3 continued*

generically by dotted lines between homologous CRM modules that promote the likelihood of CRM-promoter interactions (arrows). (**N**) Classic pairing-dependent transfection in which a functional CRM on one allele acts in-trans to promote transcription of the homologous allele. In this particular example, the functional CRM is in-cis to a loss-of-function protein coding mutation (deletion), while the functional protein coding region of the homologous chromosome is in-cis to a non-functional CRM (deletion of both sub-modules of the CRM). In this situation, all productive CRM-promoter interactions are in-trans. (**O**) A chromosomal rearrangement that disrupts pairing between homologous chromosomes blocks transvection. (**P**) When one of two redundantly acting CRM sub-modules is deleted without interfering with precise chromosomal alignment, activity can be sustained by pairing-dependent interactions between the remaining sub-modules on both homologues. (**Q**) If pairing between homologous CRMs is disrupted by deletion of one cooperatively acting sub-module on one allele and of the other sub-module on the homologous chromosome, the sum of the weakened activities of both non-interacting sub-modules being below the required threshold for sustaining gene function. (**R**) Similarly, if pairing of homologues is disrupted by a chromosomal inversion, then the two remaining isolated sub-modules do not surpass a necessary threshold for sustaining functionally sufficient transcription of the gene. For this model, we hypothesize that generic interactions between CRM modules on homologous chromosomes may modulate CRM function. Such interactions could be sustained by a variety of potential mechanisms including (1) stabilizing protein complexes acting in cis; (2) interactions between non-coding RNAs generated from one CRM and interacting with proteins, RNA, or DNA on the opposing homolog, (3) coordinated alterations in chromatin dependent on steric access to identical chromosomal sequences.
DOI: https://doi.org/10.7554/eLife.30281.007

The following source data and figure supplements are available for figure 3:

**Figure supplement 1.** Range of *riΔ3* mutant phenotypes based on chromosome pairing.
DOI: https://doi.org/10.7554/eLife.30281.008

**Figure supplement 1—source data 1.** Raw Data for *Figure 3—figure supplement 1*.
DOI: https://doi.org/10.7554/eLife.30281.009

## Replacement of the *D. melanogaster kni* L2-CRM with orthologous sequences from different fly species

As mentioned above, we identified three highly conserved sequence islands (termed 'ID' for identity) within the L2 CRM that displayed virtually no nucleotide variation across the entire Drosophilid clan (*Figure 1A* magenta, red, and green blocks ID1, ID2, ID3; their location in the CRM in *Figure 5A*, and IDs sequence alignment in *Figure 5—figure supplement 1*). We used these short conserved ID sequences alone or in combination to identify homologous regulatory regions from yet more distantly related flies in the broader Schizophora group (*Wiegmann et al., 2011*) including the housefly (*Musca domestica*) and the Medfly (*Ceratitis capitata*). Notably in these later species, sequence conservation in the candidate L2-CRMs compared to *D. mel.* was restricted to the vicinity of the ID islands (*Figure 5A*, *Figure 5—figure supplement 1*). In lieu of the guidance that multi-sequence alignments provided within the Drosophilid clade to identify the likely bounds of the L2-CRM in other species, we included ~200–400 bp before the ID-1 sequence and ~600–1000 bp after the ID-3 sequence for *M. domestica* and *C. capitata*, respectively, with a goal of also retaining sequences conserved within the various clades. We inserted candidate orthologous genomic fragments from donor species into a slightly modified *riCC-AB* CopyCat vector (*Figure 5B*). This arrangement permits cargo insertion and, if desired, CRE-mediated excision of non-cargo sequences, which also results in loss of the active genetic potential of the transgenic element (see *Figure 5—figure supplement 2*).

We tested CRM replacements for three fly species in these proof-of-principle experiments (*Figure 5C*, compare with *Figure 5—figure supplement 2* for control experiments in which non-CRM sequences were deleted by CRE-mediated recombination), one distantly related species within the Drosophilid group (*Drosophila grimshawi*) ~45 Mya, and two Schizophora (*M. domestica* (*Scott et al., 2014*) and *C. capitata* [*Papanicolaou et al., 2016*]) spanning evolutionary divergence periods of ~60–75 Mya. *D. grimshawi* is significantly larger than *D. mel.*, but has a very similar vein pattern, while in both *M. domestica* and *C. capitata* the position of the L2 vein is shifted anteriorly relative to other elements of the wing (note that *M. domestica* is also much larger than *D. mel.*) (*Figure 5C*). As expected, a control strain homozygous for a CopyCat element that deletes and restores the original *D. mel.* L2-CRM sequences (*riCC-D.mel.*) exhibited full rescue of the L2 vein leading to Kni expression in third instar discs that was indistinguishable from wild type (*Figure 6C,D*; compare to 6A, B). Similarly, flies homozygous for the *D. grimshawi* L2-CRM (*riCC-D.grim.*) exhibited a complete L2 vein in approximately the correct position and strong Kni expression in the expected

pattern in wing discs (*Figure 6E,F*), consistent with the similar relative vein spacing patterns within the Drosophilid clade.

We expected that at greater phylogenetic distances beyond the Drosophilid clade that candidate L2-CRM replacements in *D. mel.* might fail to support Kni expression or rescue of the L2 vein. What we had not anticipated, however, was that the *riCC-M.dom.* or *riCC-C.cap.* L2-CRM replacements for *M. domestica* and *C. capitata* would result in shifts of the positions of rescued L2-vein segments. Strains homozygous for the *riCC-M.dom.* or *riCC-C.cap.* replacement alleles displayed complete (*riCC-M.dom.*) or partially (*riCC-C.cap.*) restored L2 veins that were substantially displaced toward the anterior of the wing relative to the endogenous *D. mel.* L2 vein (or the control strains homozygous for the *riCC-D.mel.* element) (*Figure 5C*). In the case of *riCC-M.dom.* wings, the full vein coursed in a much sharper anterior angle from its normal point of branching from the L3 primordium, resulting in the rescued vein intersecting the wing margin in a position substantially proximal to that of the wild-type L2 vein (*Figure 5C*, see figure legend for quantitation). Presaging the anterior displacement of the rescued L2 vein in adults, expression of Kni in third instar wing discs was significantly broadened in *riCC-M.dom.* discs (*Figure 6G,H,N-P*) and pre-pupal wings (*Figure 2—figure supplement 1S–U*), extending further to the anterior than in wild-type wing primordia, but sharing the same posterior limit abutting the Spalt expression domain as in wild-type discs [*Figure 6K–M*; *Figure 2—figure supplement 1G–H*]). Kni expression in wild-type *M.dom.* discs is similarly broadened, suggesting that the pattern observed in the *riCC-M.dom.* replacement faithfully reproduces the endogenous pattern in its species of origin (*Figure 6—figure supplement 1B,C*). The exclusion of *riCC-M.dom.* Kni expression from the central domain of high level Spalt expression suggests that the *M.dom.* CRM, like that of *D.mel.*, is subject to Spalt-mediated repression. The net anterior displacement of the adult L2 vein is similar to that resulting from low-level ubiquitous expression of *kni* in an *ri1* mutant background (*Lunde et al., 1998*) and may reflect subsequent lateral inhibitory interactions during pupal stages that restrict vein formation to narrow stripes within a broader pro-vein domain (*Biehs et al., 1998*; *Sturtevant and Bier, 1995*). Vein rescue was only partially complete for homozygotes carrying the *riCC-C.cap* CRM replacement, which consistently had a rescued central L2 vein segment running parallel to the margin at a much reduced distance than observed for the wild-type *D. mel.* vein (*Figure 5C*, see figure legend for quantitation). Since little, if any, Kni expression is restored in *riCC-C.cap* wing discs (*Figure 6I,J*) or pre-pupal wings (*Figure 2—figure supplement 1F*), it is likely that the partial vein rescue observed with this construct reflects activity of this CRM at a later developmental stage. This hypothesized *kni* expression in the L2 primordium is consistent with the observation that the endogenous *kni* and *salm* expression patterns in *C. capitata* (*Figure 6—figure supplement 1*) and *D. mel.* are similar. As the observed anterior shifts of L2 veins observed in both the *riCC-M.dom.* and *riCC-C.cap* CRM replacements reflect the relative positions of L2 veins in *M. domestica* and *C. capitata,* we tentatively conclude that features of the genetic information for proper positioning of the L2 veins in these species reside within *kni* cis-regulatory sequences themselves and that these L2-CRMs are not merely executing patterning decisions imposed on them by more upstream elements of the wing gene regulatory network.

## Discussion

The advent of efficient high fidelity genome editing has transformed all fields of biology including the functional analysis of cis-regulatory elements impinging on key nodes in gene regulatory networks. In this study, we illustrate several important advantages of performing *in locus* CRM analysis as an alternative to traditional CRM-reporter constructs. First, altering CRM sequences in their native chromosomal environment can reveal subtle or context-dependent forms of regulation that are lost or obscured when performing analysis of CRM-reporter genes inserted into ectopic chromosomal sites. For example, the novel L2-CRM alleles we generated at the *kni* locus revealed previously unappreciated trans-interactions that may reflect chromosome-pairing-dependent CRM functions, as well as evidence for a proximo-distal patterning input to L2 specification. Second, phenotypic readouts can be far more sensitive discriminators of CRM function than assessments of gene expression as they represent the function of the CRM compounded over time. Third, the combination of reading out endogenous gene expression patterns and their resulting morphological phenotypes can identify distinguishing features of mutants that neither measure alone could resolve. Finally, we demonstrate super-Mendelian inheritance of active genetic elements (e.g. CopyCat vectors) which have the

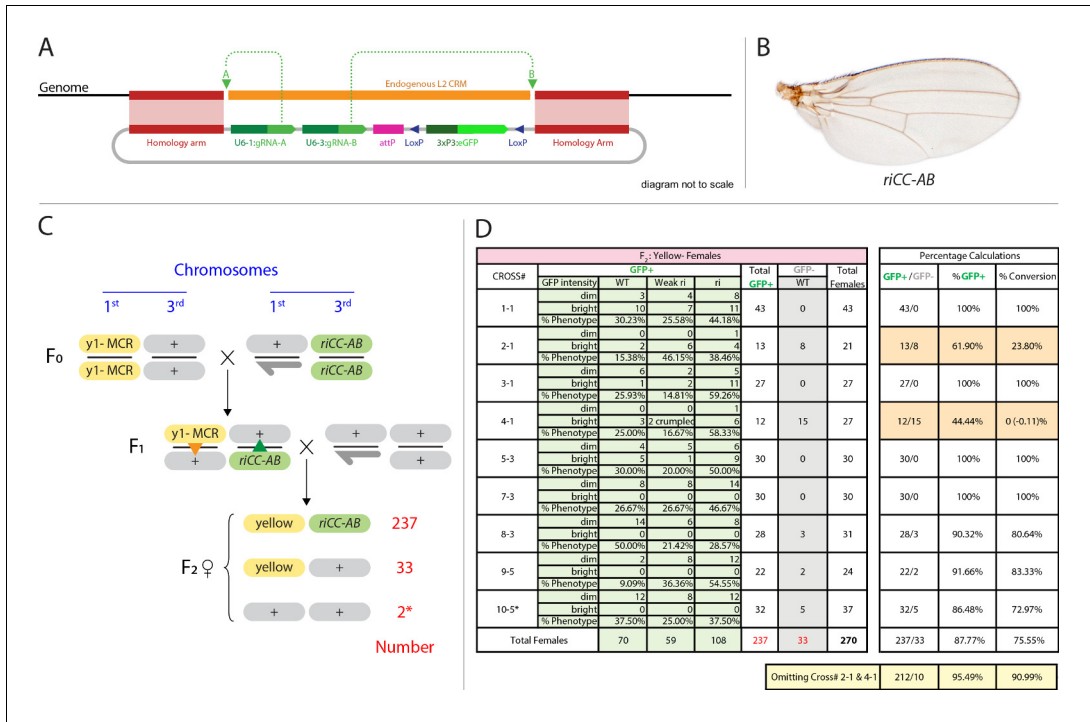

**Figure 4.** Efficient super-Mendelian transmission of a double-cut CopyCat element designed to delete the L2 CRM. (**A**) Diagram of the transgenesis vector for creating the *riCCAB* double-cut CopyCat (CC) element, the full nomenclature for which (*kni*^ri<CC|gRNA-A,gRNA-B|3xP3-EGFP>), is denoted according the following convention that we propose for representing active genetic elements in this and future studies. We have previously outlined features of a variety of active genetic elements including: MCRs, ERACR type reversal drives, CopyCat (CC) elements, and Trans-complementing drives (TC) (***Gantz and Bier, 2016***). Our proposed nomenclature convention for active genetic elements uses '<>" to denote that the element is 'active': *gene locus*^<construct type (i.e. MCR, ERACR, CC, TC)| gRNAs| dominant marker| other 5' or 3' cargo>. (**B**) Homozygous *riCC-AB* adult wing (compare to ***Figure 1C***). (**C**) Crossing scheme: Homozygous y1-MCR; +/+females were crossed to homozygous +/Y; *riCC-AB/riCC-AB* males and resulting F1 female progeny of the genotype y1-MCR/+; *riCCAB*/+were then crossed to w-/Y; +/+males. F2 female progeny were then scored for inheritance of the y1-MCR element (assessed by a full body *yellow*- phenotype) as well as for the EGFP-marked *riCC-AB* element. In addition, F2 females were scored for the presence (weak or strong) or absence of an *ri* phenotype. (**D**) Table of nine crosses showing the fraction of *y*- females which inherited the GFP marker (number and %GFP+) and among those that did, the numbers and percentages of those with wild-type, weak *ri*, strong *ri, or* total with *ri* wing phenotypes, and percent of total progeny with *ri* phenotypes. Results from the table shown in panel (**D**) can be summarized as follows: In a total of nine separate crosses, transmission of the y1-MCR was observed in 270/272 female progeny (=99.3%). Two individuals derived from cross 10–5* were y+. In four of the crosses the *riCC-AB* element was transmitted to 100% of the progeny, in three crosses to an average of 89% of progeny, and in two crosses only Mendelian inheritance was observed (note: nearly identical percentages were observed for *riCC-AB* transmission in males from the same set of 9 crosses). Note that there is no correlation between the % GFP +and the % ri (weak +strong). In particular, approximately 70% of progeny exhibited some degree of ri phenotype in all four crosses that transmitted the *riCCAB* to 100% of their progeny as well as in the two crosses that only transmitted the *riCCAB* to ~50% of their progeny. These results suggest that the germline and somatic activities of this active genetic element seem separable events.
DOI: https://doi.org/10.7554/eLife.30281.010

The following source data is available for figure 4:

**Source data 1.** Raw Data for *Figure 4*.
DOI: https://doi.org/10.7554/eLife.30281.011

potential to greatly accelerate the assembly of complex genetic stocks thereby facilitating a new era of synthetic biology (***Gratz et al., 2013***).

## Novel insights into L2-CRM function provided by *in locus* analysis

The ability to generate targeted deletions and combinations of such lesions while recovering a myriad of imprecise lesions as a byproduct of CRISPR mutagenesis strategies, fundamentally transforms the process by which CRM function can be analyzed. In this study, we highlight three fundamentally new insights and attendant questions into the activity of a CRM that we had thought we already understood quite well.

(1) Do CRMs on homologous chromosomes cooperate in a pairing-dependent fashion? The vastly different phenotypes of the *riΔ3* deletion (and to a lesser extent *riΔ2* as well) when homozygous versus in-trans to the larger full CRM *riAB* deletion was the first major surprise of this current study. In the case of other previously identified partial CRM loss-of-function mutations such as *ri1* or *ri53j*, the phenotypes of homozygotes are only modestly increased when placed in trans to *riAB* as is typical when a hypermorphic allele is placed over a deletion. The *riΔ3/riΔ3* phenotype, however, is nearly wild-type, while that of *riΔ3/riAB* trans-heterozygotes results in significant vein loss. This type of trans-chromosomal interaction, which we refer to as CRM-synergy, is reminiscent of, the phenomenon of transvection as described originally by Ed Lewis (*Lewis, 1954*). In the case of transvection, a regulatory entity (CRM in modern parlance) on one chromosome promotes expression of a trait (activates transcription) on the other chromosome (*Figure 3N*), and this trans-interaction between the CRM and basal promoter is abrogated by chromosomal inversions that interfere with chromosomal pairing (*Figure 3O*).

In contrast to transvection, for CRM-synergy we hypothesize that there are interactions between the two CRMs of different chromosomes that mutually reinforce the abilities of both CRM elements to engage either basal promoter. One possible explanation for CRM-synergy in the case of the *riΔ2* or *riΔ3* deletions is that two mutually reinforcing CRM sub-complexes could form on sequences present in either region deleted in *riΔ2* or *riΔ3*, and that these complexes normally act in concert to provide full CRM activity (*Figure 3M*). Perhaps when either of the *riΔ2* or *riΔ3* deletions are homozygous, the remaining complexes present on the other remaining submodule can sustain CRM function as long as those elements on the two homologous chromosomes are well paired and can benefit from the trans-CRM interaction. When they are placed over the larger deletion, however, this pairing breaks down since the one remaining submodule can no longer interact productively with its homologous partner. This hypothesis is further supported by the observation that the phenotype of *riΔ3/ri1* (*Figure 3B*, modeled in *Figure 3P*) is aggravated by interruption of chromosome pairing by the rearranged *TM3, ri1* balancer chromosome (*Figure 3F*, modeled in *Figure 3R*) and that *riΔ2/riΔ3* trans-heterozygotes display a substantial vein-loss phenotype (*Figure 3C*, modeled in *Figure 3Q*). It is also noteworthy that there is only a modest vein loss phenotype manifested in *riΔ3/ri1* flies despite the fact that the *riΔ3/riAB* and *ri1/riAB* phenotypes are comparable. This effect may arise from the fact that deleted sequences in *ri1* overlap with those missing in *riΔ3*. Consistent with this hypothesis, the *riΔ3/ri53j* phenotype is substantially stronger than that of *riΔ3/ri1* despite the fact that the *ri53j/ri53j* phenotype is typically less severe than that of *ri1/ri1*. Since *ri53j* is a point mutant allele that is covered by the *riΔ3* deletion, alignment of homologous chromosomes in *riΔ3/ri53j* trans-heterozygotes would likely be disrupted as compared to *riΔ3/riΔ3* homozygotes that entirely lack the mutated nucleotide in *ri53j* and surrounding sequence. Further analysis will be required to determine the nature of the hypothesized pairing-dependent interaction and to exclude other potential explanations such as 1) the potential creation of novel transcription factor binding sites at the junction points of particular deletions which could then alter vein phenotypes by a neomorphic mechanism or 2) the *ri53j* mutation disrupts binding of an activator that normally acts by overcoming the formation of an inhibitory complex on adjacent sequences that are deleted in the *riΔ3* mutant (these most obvious alternative explanations, however, do not readily account for the inversion-dependent differences in the *riΔ3/ri1* phenotype nor the failure of *riΔ3* and *riΔ2* deletions to complement). Another question that remains to be addressed is why the *ri53j* point mutant allele has a stronger homozygous phenotype than the *riΔ3* deletion that covers it. One possibility to explore is that the *riΔ3* deletes both necessary activator sites (e.g. the *ri53j* site) and repressor sites leading to a weaker homozygous phenotype than deletion of just the activator sites.

(2) Split, cooperative, negative regulatory elements in the L2-CRM. In-locus deletion of sequential regions of the full L2-CRM suggested that two separate regions of the enhancer might contribute to repression of *kni* expression in peripheral regions of the wing primordium. Remarkably, when these

two deletions were combined, a highly synergistic ectopic vein phenotype was observed. The ability to assay the effects of these deletions directly on gene function was a key feature of the experiments that reveal this strong cooperative phenotypic interaction. This observation motivates future studies to understand the mechanism by which the separated CRM sequences might interact. Do they assemble similar protein complexes that act in a redundant fashion? Might the two domains bind different transcription factor complexes but loop in some way to compete with activators binding to central regions of the CRM or to block a pairing-dependent form of trans-chromosomal cooperative activation? Another hypothesis to consider is that sequences in the *riΔ1* region include not only sites for repressors, but also for inputs that limit anterior *kni* expression during early prepupal stages. Recall that the double *riΔ1+Δ4* and single *riΔ4* mutants exhibit comparable levels of ectopic Kni expression in larval stages (*Figure 5H,L*), but that the zone of strong ectopic Kni expression seems narrowed during pre-pupal stages in *riΔ1+Δ4* relative to *riΔ4* (*Figure 2—figure supplement 1O,R*). These observations might help explain why the *riΔ+,Δ4* double mutant has a more extreme ectopic vein phenotype than the *riΔ4* single mutant since previous studies have shown that high-level ubiquitous *kni* expression eliminates expression of both vein and intervein markers, possibly via overly exuberant induction of lateral inhibitory processes. According to this seemingly paradoxical hypothesis, the narrower zone of high level Kni expression observed the *riΔ1+Δ4* double mutant may be less subject to vein elimination by such lateral inhibitory influences than the *riΔ4* single mutant with persistent high level broad ectopic Kni expression throughout the anterior domain. Further analysis will be required to test this hypothesis. Also, one puzzle that remains unanswered by the current analysis is how the Spalt locus might negatively regulate CRM function in central regions of the wing as we failed to recover any mutations that induced ectopic veins in this territory. Unfortunately, the DNA-binding sites for Spalt proteins (Salm and Salr), if any, in the L2-CRM, are among the few transcription factors for which a well-defined binding site consensus remains to be determined (*de Celis and Ostale, 2017*). Perhaps, the failure to recover mutants in all our different experiments that induce ectopic veins in central regions of the wing reflects an interspersion of Sal repressor sites with activator sequences (e.g. known functionally important Scalloped sites)? Further refined deletion analysis of activator domains should help address this question as well as studies to identify *bona fide* Spalt protein binding sites. Sequence comparisons with the highly diverged L2-CRMs from *M. domestica* and *C. capitata* may prove helpful in this effort, as the exclusion of Kni expression from the central Spalt expression domain suggests that at least the *riCC-M.dom.* CRM retains sensitivity to Spalt-mediated repression, although this effect could be indirect. We note that these new features and models of *kni* regulation derive from a combination of morphological and *in locus* gene expression data that were not revealed previously based solely on analysis of traditional L2-CRM-reporter gene fusion constructs (e.g. [*Martín et al., 2017*; *Lunde et al., 2003*]).

(3) Novel unexpected phenotypes. Approximately half of CRISPR-induced mutations we recovered were the targeted mutations we set out to isolate. The remaining fortuitously generated lesions included imprecise indels at one or both of the targeted cleavage sites or, in some cases, deletions or duplications of short sequences at a distance from the cleavage sites that may reflect an error-prone form of HDR repair based on the presence of micro-homologies (sometimes referred to as micro-homology-mediated end joining or MMEJ). In some cases, these unanticipated mutations generated surprising phenotypes, including L2 deletions biased to proximal or distal regions of the vein, dramatically different phenotypes of deletions resulting from minor differences in endpoints, and in one case the addition of a single base in a non-conserved region of the CRM resulting in a vein-loss phenotype. While the basis for these various unexpected phenotypes remains to be determined, the recovery of a diverse array of ancillary mutations accompanying targeted CRISPR-based mutagenesis offers a potential treasure trove for gaining unbiased insights into CRM function.

## Inter-species CRM replacements using active CopyCat elements

We previously demonstrated a new CRISPR-based method referred to as the Mutagenic Chain Reaction (MCR) that results in efficient copying of genetic elements to the homologous chromosome during meiosis resulting in strong gene drive (*Gantz and Bier, 2015*). Similar gene-drive systems have also been shown to transmit efficiently to offspring in mosquitoes (*Gantz et al., 2015*; *Hammond et al., 2016*) and to copy efficiently in diploid yeast where such mutations can be transmitted subsequently during meiosis (*DiCarlo et al., 2015*). Based on these initial studies, we proposed the design of a new type of transgenesis vector that we refer to as CopyCat elements

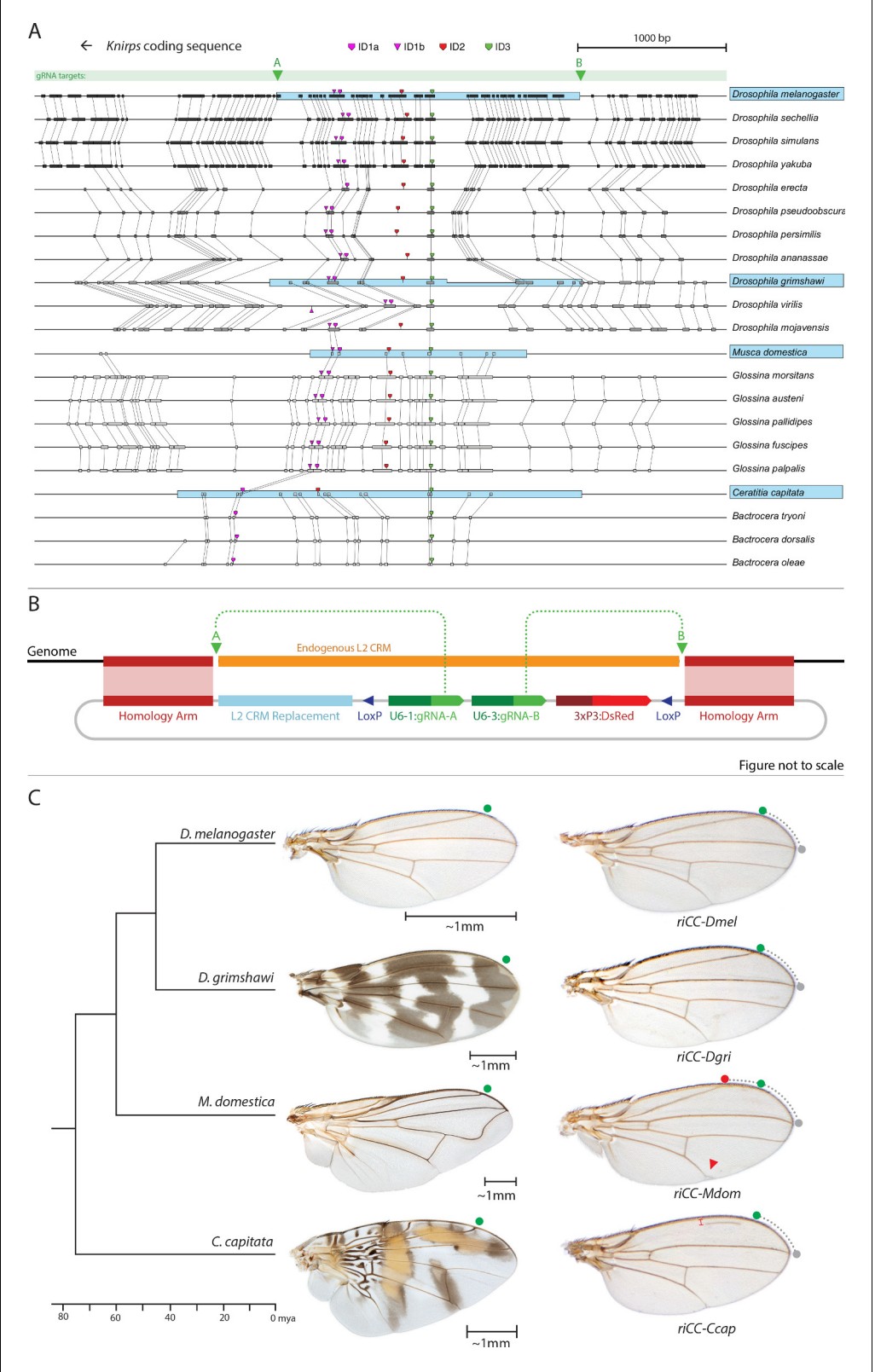

**Figure 5.** Replacement of the *D. melanogaster* L2-CRM with orthologous sequences from other Dipterans. (**A**) Comparisons of candidate L2-CRM sequences across dipterans tested for replacement. Vertical lines connect boxes (black grey) indicating conserved identical DNA sequences. Hyper-conserved identity islands (ID sequences - shown with different colored symbols) were used to identify the L2-CMR candidate sequences (light blue boxes)

*Figure 5 continued on next page*

*Figure 5 continued*

in whole genome sequence BLASTS. Drosophilid species names within the different clades shown in the alignment that were tested for CRM function are boxed in blue. (B) Map of CopyCat CRM replacement vector: Diagram of plasmid used for transformation with CopyCat replacements. Elements depicted from left to right are: Homology arms (magenta boxes); replacement CRM sequences (light blue box - not drawn to scale); paired LoxP sites allowing deletion of non-replacement sequences if desired (dark blue triangle); gRNAs A and B (green arrows boxes) expressed under the control of different U6 promoters (dark green boxes); and the DsRed dominant eye color marker (red arrow) expressed under the control of the 3XP3 eye-specific promoter (maroon box) followed by 3′ polyadenylation sequences derived from SV40 (not shown in diagram). Endogenous L2 CRM shown in orange (not drawn to scale with regard to CRM replacements - see panel A for proportional comparisons of CRM sizes and endpoints). (C) Replacement of the *D. mel.* L2-CRM with candidate orthologous sequences from other Dipterans. Approximate phylogenetic tree of Dipteran species tested on left, examples of wings from these species in the center with relative size bars and green circles indicating respective positions of the L2 vein in each species. Typical examples of vein replacement phenotypes which were highly reproducible in all cases. Note that the *riCC-Dmel* (full notation: $kni^{ri<CC|gRNA-A, \, gRNA-B|3xP3-DsRed|5'-Dmel-L2-CRM>}$) and *riCC-Dgri* ($kni^{ri<CC|gRNA-A,gRNA-B|3xP3-DsRed|5'-Dgri-L2-CRM>}$) replacements provide rescue of the L2 vein that is indistinguishable from the wild-type position, shown with green circles marking the wild-type position of L2 compared to L3 - dotted arc connecting to grey circles at L3 position. In *riCC-Mdom* ($kni^{ri< CC|gRNA-A, \, gRNA-B|3xP3-DsRed|5'-Mdom-L2-CRM>}$) wings, the L2 vein is also fully rescued but intersects the wing margin in a much anteriorly displaced position (red circle). This anterior displacement phenotype was quantitated in N = 20 wings by three separate measures: (1) ratio of the length of L2/L3 veins: average = 0.781 ± 0.006 for WT versus 0.637 ± 0.016 for *riCC-Mdom*, p<$1.2\times10^{-18}$; (2) the angle between the L2 and L3 veins diverging from their proximal intersection point: average = 13.63 ± 0.58 degrees for WT versus 18.64 ± 0.58 degrees for *riCC-Mdom*, p<$2.0\times10^{-17}$; and (3) the number of thick innervated bristles forming anterior to the point of intersection of the L2 vein with the margin: average = 11.15 ± 0.9 bristles for WT versus 28.9 ± 1.8 bristles for *riCC-Mdom*, p<$2.8\times10^{-20}$. In *riCC-Ccap* ($kni^{ri< CC|gRNA-A,gRNA-B|3xP3-DsRed|5'-Ccap-L2-CRM>}$) wings, only rescue of a partial vein segment is observed (N = 20 wings; average ratio of L2 vein segment length to that of L3 = 0.136 ± 0.03 for *riCC-Ccap* versus 0.781 ± 0.006 for WT, p<$7.7\times10^{-27}$) although its occurrence and placement are highly penetrant phenotypes. This rescued vein segment forms much closer to the margin than does the normal L2 vein in *D. mel*. This effect was quantitated by measuring the relative distance of the rescued vein segment from the margin at a defined PD position (arrow) relative to the distance of the L2 vein in WT at that same PD position: (N = 20 wings; average distance in *riCC-Ccap* rescue = 0.058 ± 0.004 mm versus 0.091 ± 0.005 mm for WT, p<$6.9\times10^{-15}$). Note that in both *M. domestica* and *C. capitata* wings, the L2 vein forms in a relatively more anterior position than it does in *D. mel.* or *D. grimshawi.*.

DOI: https://doi.org/10.7554/eLife.30281.012

The following source data and figure supplements are available for figure 5:

**Source data 1.** Raw data for quantifications of L2 vein phenotypes in CopyCat transformants.
DOI: https://doi.org/10.7554/eLife.30281.013
**Figure supplement 1.** Alignment of ID sequences across Schizophora.
DOI: https://doi.org/10.7554/eLife.30281.014
**Figure supplement 2.** Homozygous phenotypes of CRM-only replacements.
DOI: https://doi.org/10.7554/eLife.30281.015

(*Gantz and Bier, 2016*) that could be copied efficiently to the homologous chromosome during meiosis in a Cas9-dependent fashion thereby bypassing traditional constraints of Mendelian inheritance, such as those associated with independent chromosomal assortment or genetic linkage. CopyCat elements can contain either a single gRNA that cuts at the site of vector insertion into the genome or two gRNAs that generate a self-propagating deletion of sequences lying between the two gRNA cut sites. In this study, we demonstrate that double cut CopyCat cloning vectors function as efficiently as conceived. In addition, CopyCat vectors can be used routinely to recover site directed genomic insertion making them an attractive alternative vehicle for transgenesis. Furthermore, when CopyCat elements are in presence of a Cas9 source, they can be passed on via the germline with super-Mendelian efficiency, a feature which should substantially simplify and shorten otherwise tortuous genetic schemes needed to assemble complex arrays of transgenes. If active genetics systems can also be developed in vertebrate and plant systems, they could revolutionize genetic manipulations in an array of organisms, in which standard Mendelian constraints pose a yet greater impediment than in fruit flies.

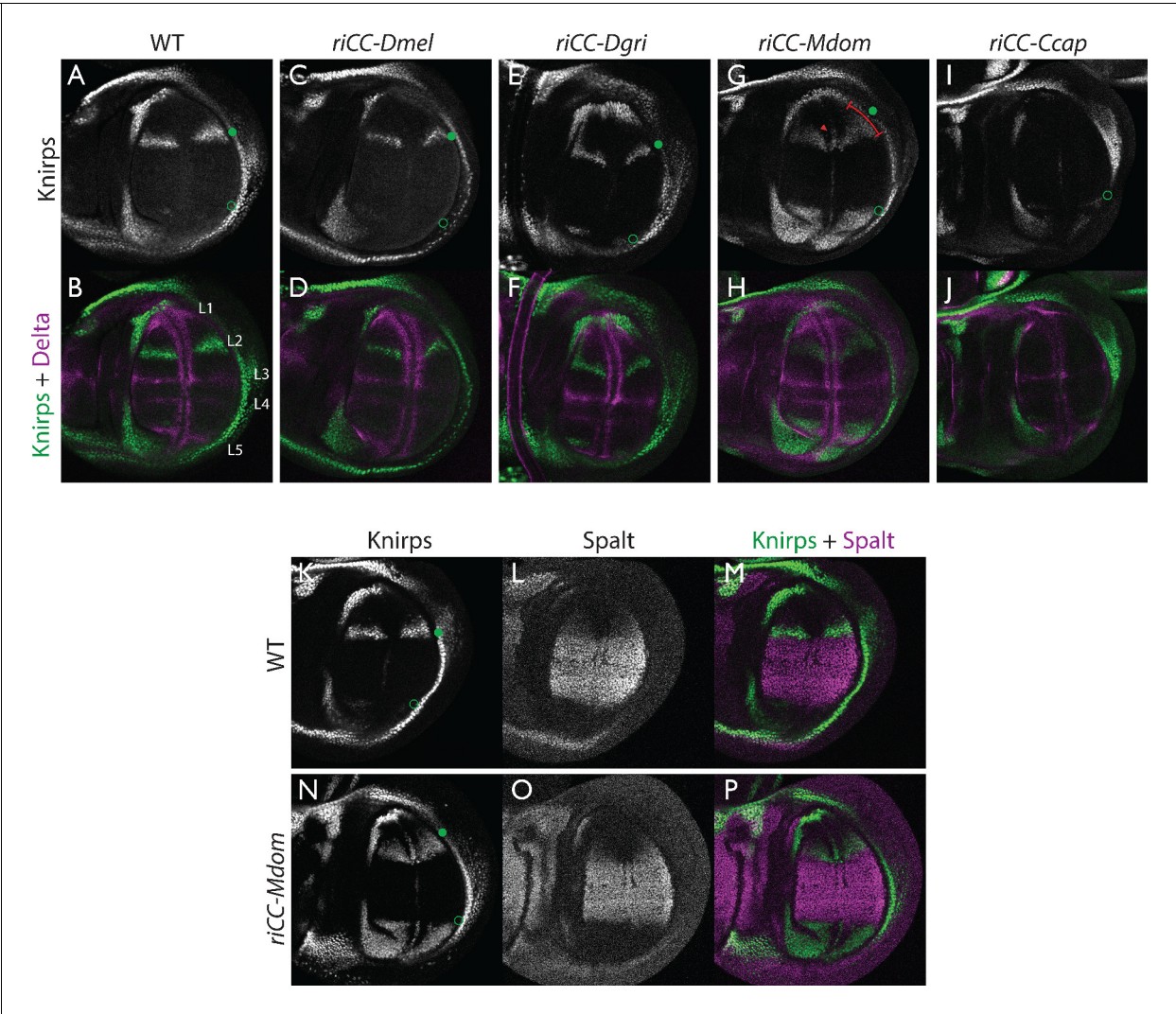

**Figure 6.** Kni expression in L2-CRM replacement lines. Double stains of third larval instar wing imaginal discs from stocks carrying homozygous CRM replacement CopyCat elements showing expression of Kni +Delta (**A–J**) or Kni +Spalt (**K–P**). Green dot indicates approximate location of the L2 primordium and open green circles indicate posterior Sal border where a faint stripe of Kni expression is detected in wild-type D. mel wings. For panels showing double stains Kni expression is in green and Delta or Spalt expression is in magenta. Delta is a pan-vein marker expressed in the L1, L3, L4, and L5 vein primordia (labeled in panel **B**) and provides a positional reference for Kni expression. Spalt labels the central region of the wing where it suppresses expression of Kni in *D. mel.* Staining in panels showing single antibody panels alone are in black and white. (**A,B**) WT wing disc - see also Spalt double labels in panels (**K-M**; **C,D**) control replacement with *D. mel.* CRM (*riCC-Dmel*); (**E,F**) replacement with *D. grimshawi* candidate L2 CRM sequences (*riCC-Dgri*); (**G,H**) replacement with *M. domestica* candidate L2 CRM sequences (*riCC-Mdom*) - see also Spalt double labels in panels (**N-P**; **I,J**) replacement with *C. capitata* candidate L2 CRM sequences (*riCC-Ccap*). (**K–M**) Wild-type wing disc stained for Kni and Spalt. Note that Kni is expressed just anterior to cell strongly expressing Spalt. (**N–P**) *riCC-Mdom* replacement disc double stained for Kni and Spalt. Note that Kni expression is broadened relative wild-type but shares the same posterior limit as in the wild-type disc. These experiments indicate that all rescued 'replacement' veins form with the same posterior border bounded by strong Spalt expression as in *D. mel.*. Kni expression in L2 primordium of *riCC-Dgri* discs is normal in position and width albeit perhaps a bit weaker than with the *D. mel.* control replacement (*riCC-Dmel*). In the case of the *riCC-Mdom*, the rescued vein is broader than in *D. mel.* extending further to the anterior. In addition, there is ectopic expression along future wing margin which is absent in *D. mel.* or the other replacements. In the case of the *riCC-Ccap* element, which provides only partial rescue of the adult L2 vein, Kni expression is virtually undetectable at this stage.

DOI: https://doi.org/10.7554/eLife.30281.016

The following figure supplement is available for figure 6:

**Figure supplement 1.** Knirps protein expression in *Musca domestica* and *knirps* (*kni*) mRNA and *spalt-major* (*salm*) mRNA expression in *Ceratitis capitata* wing imaginal discs.

DOI: https://doi.org/10.7554/eLife.30281.017

Potential caveats for the transmission of active genetic elements: The high frequency of CopyCat transmission in the experiments presented in this study suggest that this should be a generally applicable method. We noted, however, that in two of the nine crosses, transmission of the element was only Mendelian. The basis for the failure of the active copying process in such instances remains to be further evaluated, however, based on previous analysis of MCR transmission in flies (*Gantz and Bier, 2015*) and mosquitoes (*Gantz et al., 2015*; *Hammond et al., 2016*), the most likely explanation is that NHEJ repair can in some instances intervene between the time of egg fertilization and the partitioning of germ cell lineages, which takes place following seven cell divisions in the blastoderm embryo. In *Drosophila,* the frequency of such NHEJ events seems to be fairly low as revealed by propagation of the y1-MCR occurring at comparable frequencies via eggs that either are preloaded or not with a source of Cas9 (e.g. eggs from females that carry the MCR element versus eggs from wild-type females fertilized by MCR males). In contrast, in mosquitoes there is a large difference in transmission frequency between these two scenarios (98–99% chromosome conversion via males versus only 12–25% conversion via females - [*Gantz et al., 2015*]). The basis for this difference in male/female propagation between the two Dipteran species remains to be determined. However, it may be that for double cut CopyCat elements, which by their nature are more demanding of the HDR repair process than single cut MCRs, that non-HDR events are more frequent. Alternatively, the nuclease targeting of the two gRNA sites may occur asynchronously or the Cas9 enzyme may preferentially bind one gRNA thereby reducing cleavage directed by the second gRNA. In either of these scenarios, if the gRNA-A cuts before the gRNA-B, it could generate an NHEJ event at the location 'A' preventing successful HDR-driven chromosomal conversion of the CopyCat element at a subsequent cell cycle, when the gRNA-B has a chance of successfully guiding Cas9 to generate a dsDNA cleavage. Future experiments should resolve this question. Another factor to be kept in mind when using CopyCat elements as a genetic tool is that DNA repair mechanisms involved in copying these elements may be more error prone than standard cellular DNA replication (*Malkova and Haber, 2012*).

CopyCat CRM-replacement elements should accelerate evo-devo analysis of gene-regulatory networks: We also present proof-of-principle experiments showing that CopyCat elements can also carry gene sequences of interest that replace those deleted in a single genetic event. Such CopyCat elements should greatly facilitate the assembly of multiplex replacements to help identify genetic sequences from divergent species that underlie the morphological diversification. Indeed, we were surprised that replacement of just the *kni* L2-CRM with candidate homologous regulatory sequences from the housefly or Medfly led to pronounced anterior shifts of the L2 vein typical of vein placement in those donor species, suggesting that alterations in this single down-stream regulatory element are capable of mediating significant morphological changes. The nature of the salient changes in the L2-CRM remain to be investigated. Candidates for negative regulators in peripheral regions of the wing primordium that may contribute to these anterior vein shifts include the repressors Brinker and Optix/Six3 which have been implicated as part of a cross-inhibitory network involved in L2 development (*Al Khatib et al., 2017*; *Martín et al., 2017*). Intriguingly, knock-down of Optix function in anterior regions of the wing results in large anterior shifts of the L2 vein (*Al Khatib et al., 2017*; *Martín et al., 2017*), although this phenotype results at least in part from decreased proliferation of cells in the anterior domain where Optix is normally expressed (*Martín et al., 2017*). It also will be of significant interest to compare such changes that have occurred in different lineages that have yielded similar alterations in vein positions. For example, will similar or different sequences be identified as the key substrates for the anterior vein shifts we observe in *M. domestica* versus *C. capitata*? Sequencing and selective functional testing of additional species may provide estimates for how often such alternative evolutionary trajectories are taken. Also, how many CRMs of genes acting in the wing gene regulatory network would need to be replaced to transform the *D. mel.* wing into one resembling a housefly or Medfly? CopyCat-mediated CRM replacements should greatly enable studies to answer these and other frontier questions.

## The impact of genome editing approaches on cis-regulation studies

The studies presented here highlight the phenomenal impact of CRISPR-Cas9 mediated genome editing and active genetics in *Drosophila*, and by extension, most likely in many other organisms. These methods that efficiently generate targeted mutations, as well as a mix of fortuitous mutations in regulatory regions, provide unprecedented genetic leverage in dissecting non-coding DNA

functionality. For example, as described above, *in locus* analysis of CRM function based on a combination of readout of endogenous gene expression at different developmental stages and on final morphological phenotype, revealed features of *kni* regulation that were not evident from previous CRM-gene-fusion studies. These new insights include: proposed chromosome pairing dependent CRM interactions, identification and analysis of complex cooperative interactions between CRM subdomains, isolation of novel CRM mutations with unexpected phenotypes, and subtle phenotypic analysis of divergent CRM function during evolution. In addition, the comparable efficiency of CRISPR-based HDR-mediated transgenesis to that of traditional methods offers the possibility of delivering genetic cargo of interest to any desired chromosomal location. It should also be possible to develop genome-spanning collections of active CopyCat elements capable of delivering transgenic elements to a broad array of chromosomal insertion sites in a variety of sexually reproducing organisms as alternatives to Mendelian recombination-based systems. Multiple CopyCat elements could then be rapidly combined in the presence of an unlinked source of Cas9. Manipulations of this kind should significantly facilitate the import of entire gene regulatory networks from one organism into another to address important questions in evolution and promote the goals of principle-directed synthetic biology.

# Materials and methods

**Key resources table**

| Reagent type (species) or resource | Designation | Source or reference | Identifiers | Additional information |
|---|---|---|---|---|
| Antibody (guinea pig) | guinea pig anti-knirps | *Lunde et al. (1998)* | Asian Distribution Center for segmentation antibodies: Antibody #566 | diluted 1:1000 in 1x PBS with 0.1% Tween |
| Antibody (mouse) | mouse anti-engrailed | Developmental Studies Hybridoma Bank (DSHB) | RRID:AB_528194 | diluted 1:200 in 1x PBS with 0.1% Tween |
| Antibody (mouse) | mouse anti-delta | Developmental Studies Hybridoma Bank (DSHB) | RRID:AB_528224 | diluted 1:500 in 1x PBS with 0.1% Tween |
| Antibody (rabbit) | rabbit anti-spalt | *de Celis et al., 1999* | | diluted 1:1000 in 1x PBS with 0.1% Tween |
| Software | Snapgene | http://www.snapgene.com/ | RRID:SCR_015052 | |
| Software | Gene palette | http://www.genepalette.org/ | | |

gRNA constructs cloning: Cloning of the gRNA constructs was performed following protocols published online on the website (CRISPR fly Design - http://www.crisprflydesign.org/) and injection mixes were prepared according to protocols described by Gratz et al. (*Gratz et al., 2013a*). Oligos were designed with 20 nucleotides of the target genome sequence with additional 5' overhangs that are complementary to BbsI restriction sites (sense: 5' CTTC-GN$_{19}$ 3') (antisense: 5' AAAC-N$_{19}$C 3'). The oligos were synthesized de novo, then annealed in T4 ligation buffer and cloned into the pU6-BbsI-chiRNA plasmid. Following successful cloning, the gRNA plasmids were transformed into One Shot TOP10 competent cells (Invitrogen #C4040) and plated for transformant screening. For cloning single and double gRNA constructs, in this study we have used: pU6-BbsI-chiRNA which was a gift from Melissa Harrison and Kate O'Connor-Giles and Jill Wildonger (Addgene plasmid # 45946), pCFD3-dU6:3gRNA was a gift from Simon Bullock (Addgene plasmid # 49410) and pCFD4-U6:1_U6:3tandem-gRNAs was a gift from Simon Bullock (Addgene plasmid # 49411).

## Fruit fly injections

Plasmids were purified using the Qiagen Plasmid Midi kit (#12191). The gRNA plasmids were co-injected with respective 120 nucleotide ssODNs (also synthesized de novo in their entirety) that contained 60 bases homologous to the sequence flanking either side of the targeted deletion. Injection mixes were assembled with two gRNA plasmids (final concentration: 250 ng/µl each) and the donor oligo (final concentration: 100 ng/µl) in a volume of 50 µl. The mixes for *ri*Δ1-*ri*Δ4 were sent to Best

Gene Inc. for injection into their Vasa-Cas9(X) stock (BDSC #51323) while CopyCat constructs were injected into the w1118 stock (BDSC #5905) with a transient source of pHsp70-Cas9 (pHsp70-Cas9 was a gift from Melissa Harrison and Kate O'Connor-Giles and Jill Wildonger (Addgene plasmid # 45945). All constructs were fully sequenced prior to injection and after recovery in analyzed transgenic fly stocks.

## Genomic DNA preparation

injected flies were singly crossed to the *ri*AB stock, approximately 20 larvae were collected from each vial, and genomic DNA was prepared. The larvae were rinsed in deionized water and homogenized with a motorized pestle in homogenizing buffer prepared according to protocols by Steller et al. (*Steller and Pirrotta, 1986*). Genomic DNA from single adult flies were prepared according to protocols by (*Gloor et al., 1993*).

PCR screening of deletion mutants: Some crosses were screened by phenotype followed by sequencing of the *kni* CRM, while others were screened by PCR using primers that would anneal to the sequence flanking the target deletion. In the case of deletions that did not display obvious overt phenotypes across the AB deletion, DNA was prepared from 20 third instar larvae from each vial of single injected F0 flies crossed to the riAB deletion as summarized above, PCR amplified with appropriate primers to detect the expected deletion, and analyzed by gel electrophoresis. Crosses giving the expected deletion band were kept and individual F1 progeny from those crosses were crossed again to the riAB deletion and re-tested. F2 flies again testing positive were then crossed to a TM3 balancer stock, and flies lacking an overt L2 phenotype (i.e. the non-AB/TM3 flies) were crossed to each other to establish a stock. Stocks homozygous for the desired deletion were then selected based on the absence of the TM3 balancer. Other primers used to screen for additional deletion events were designed with sequences lying outside of the deletion sequence but inside of the kni enhancer (AB fragment) in order to amplify the chromosome containing the newly generated deletions. PCR reactions were assembled with Phusion High-fidelity polymerase from NEB (#M0530S) and the genomic DNA previously prepared from collected larvae. PCR products were purified using the QIAquick PCR purification kit (#28104) before sequencing.

## Active genetic crosses

All crosses using active genetics were performed in accordance to an Institutional Biosafety Committee (IBC) approved protocol in a secure ACL2 insectary as previously described (*Gratz et al., 2013*) consistent with currently suggested guidelines for laboratory confinement of gene drive systems (*Hammond et al., 2016*; *DiCarlo et al., 2015*).

## CRE-mediated deletion of non-CRM CopyCat sequences

Non-CRM sequences flanked by Lox-P sites carried on the CRM-replacement CopyCat vectors were deleted by crossing the DsRed +CopyCat elements to a ubiquitous source of CRE-recombinase (BDSC #851). The resulting F1 progeny displayed a loss of the DsRed marker as trans-heterozygotes indicating early developmental CRE-mediated deletion of the cassette resulting in somatic as well as germline deletion of the DsRed cassette. These F1 flies were crossed to balancer stocks and isogenic lines were established from individual DsRed- F2 progeny. Molecular analysis of these deletion stocks revealed the clean predicted deletions of the DsRed cassette. Chromosomes carrying these unmarked elements were homozygosed by elimination of balancer chromosomes. Homozygous CRM-only replacements were analyzed for their phenotypes (*Figure 5—figure supplement 1*), which displayed little if any differences from their DsRed +parent lines (*Figure 5*).

### Wing dissection and mounting

*Drosophila* wings were dissected in 90% ethanol and mounted in 100% Canada balsam.

### Antibody staining of imaginal discs

3rd instar larvae were dissected on ice in 0.1% Tween-PBS and fixed with 2% PFA in Brower buffer for 1 hr at 4°C. Discs were stained in a mix of 1:1000 guinea pig anti-Knirps, 1:200 mouse anti-Engrailed, 1:500 mouse anti-Delta, 1:1000 rabbit anti-Spalt overnight at 4°C. Samples were mounted

in Slowfade diamond anti-fade mountant (#S36963) and imaged on a Leica SP5 confocal microscope.

## Acknowledgements

We thank Bill McGinnis for the original suggestion of creating active genetic cloning elements, Nico Posnien at the University of Göttingen for kindly pointing us to unpublished data from Natalia Siomava were generated during her PhD graduate studies in the laboratory of Ernst Wimmer at the University of Göttingen, Giuseppe Saccone for providing *C. capitata* flies for wing mounting preparation, Al Handler for providing *C. capitata* DNA samples, Alec Gerry for *M. domestica* DNA samples and flies for wing mounting preparation, Bill McGinnis, Steve Wasserman, Joseph Vinetz, Scott Rifkin, Marty Yanofsky, Kim Cooper, Hannah Grunwald, Omar Akbari, Giuseppe Saccone and members of the Bier lab for help comments on the manuscript. These studies were supported by NIH grant 1R01GM117321 to EB, the Paul G. Allen Frontiers Group Distinguished Investigator Award to EB, and a gift from Sarah Sandell and Michael Marshall to EB, NIH 1DP5OD023098 to VMG, and NIH NS047101 to the UCSD Imaging Core.

## Additional information

### Competing interests

Valentino Matteo Gantz, Ethan Bier: Founder of Synbal, Inc. and Agragene, Inc. The other authors declare that no competing interests exist.

### Funding

| Funder | Grant reference number | Author |
| --- | --- | --- |
| National Institutes of Health | 1R01GM117321 | Ethan Bier |
| Paul G. Allen Frontiers Group | Distinguished Investigator Award | Ethan Bier |
| National Institutes of Health | 1DP5OD023098 | Valentino Matteo Gantz |

The funders had no role in study design, data collection, and interpretation, or the decision to submit the work for publication.

### Author contributions

Xiang-Ru Shannon Xu, Conceptualization, Data curation, Formal analysis, Investigation, Methodology, Writing—review and editing; Valentino Matteo Gantz, Conceptualization, Data curation, Supervision, Funding acquisition, Investigation, Methodology, Project administration, Writing—review and editing; Natalia Siomava, Investigation, Contributed to this work by providing data and interpretation of the Ceratitis capitata wing disc in situ hybridization displayed in Figure 6 - Figure Supplement 1; Ethan Bier, Conceptualization, Supervision, Funding acquisition, Writing—original draft, Writing—review and editing

### Author ORCIDs

Xiang-Ru Shannon Xu http://orcid.org/0000-0002-2781-9767
Valentino Matteo Gantz http://orcid.org/0000-0003-2453-0711
Ethan Bier http://orcid.org/0000-0002-2892-3005

### Decision letter and Author response

Decision letter https://doi.org/10.7554/eLife.30281.020
Author response https://doi.org/10.7554/eLife.30281.021

## Additional files

**Supplementary files**
• Transparent reporting form
DOI: https://doi.org/10.7554/eLife.30281.018

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
