## [Decision Letter]

Thank you for submitting your article "Active genetics-based trans-species replacement of the endogenous *Drosophila* kni-L2 CRM reveals unexpected complexity" for consideration by *eLife*. Your article has been reviewed by three peer reviewers, one of whom is a member of our Board of Reviewing Editors, and the evaluation has been overseen by Detlef Weigel as the Senior Editor. The following individual involved in review of your submission has agreed to reveal his identity: Yoshinori Tomoyasu (Reviewer #2).

The reviewers have discussed the reviews with one another and the Reviewing Editor has drafted this decision to help you prepare a revised submission.

Summary:

In this manuscript, the authors present a detailed analysis of a regulatory element that regulates knirps expression in the developing *Drosophila* wing and ultimately controls the formation of the L2 wing vein of the adult. The paper makes use of both CRISPR/Cas9 for mutagenesis of the element as well as for the replacement of the enhancer with sequences from other fly species using so-called CopyCat vectors. From a technical viewpoint, the paper is an important contribution in establishing very efficient and accurate methods for swapping in new regulatory elements within a gene, and this approach holds great promise for the entire evo-devo field. On top of this technical advance, the paper presents several new and important findings using these methods. First, the work better defines the sequences important to the function of the cis-regulatory module by generating a wide variety of new mutant alleles. Second, the authors show evidence for a pairing-dependent process that governs the activity of this regulatory module. Third, the authors achieve a clean swap of the normal *D. melanogaster* CRM with the CRM from three other fly species, which leads to changes in the L2 vein that mimics the phenotypes seen in the other fly species. Clearly, there is quite a bit more detailed analysis that could be done in the future, and the paper lays the groundwork for what will be an excellent system for future analyses of cis-regulatory function and evolution.

Essential revisions:

1) All three reviewers have agreed that the paper is more appropriate as a "Tools and Resources" paper. While there are some clear biological insights, these are not followed up in sufficient detail to lead to a major advance. However, the technical aspects will be useful to others, and the work generated here will almost certainly allow these authors and others to advance this particular system and possibly create significant breakthroughs in our understanding of the mechanism at work to set the position of the L2 vein and how it has evolved. The authors should revise the paper with this in mind.

2) The authors should provide some quantitation on the shift of the position and length of the L2 vein after the enhancer swaps (from other Dipteran species). In particular, the authors should quantify the shift in L2 position seen with the *M. domestica* CRM (distance of shift after normalizing for variation in wing size) and shortening of L2 seen with the C. capitata CRM (length of L2 relative to total wing length).

3) The authors should provide some data on the expression of *kni* and *salm* in the wing disks of *M. domestica* and *C. capitata* to assess if indeed the pattern of these genes suggests general similarities in the mechanisms of vein placement, and to assess if the pattern seen when the respective CRMs are placed in *D. melanogaster* is reflective of the normal pattern of expression for kni in the source species.

---

## [Author Response]

Essential revisions:1) All three reviewers have agreed that the paper is more appropriate as a "Tools and Resources" paper. While there are some clear biological insights, these are not followed up in sufficient detail to lead to a major advance. However, the technical aspects will be useful to others, and the work generated here will almost certainly allow these authors and others to advance this particular system and possibly create significant breakthroughs in our understanding of the mechanism at work to set the position of the L2 vein and how it has evolved. The authors should revise the paper with this in mind.

We have revised the manuscript to emphasize the technical aspects of the technology where possible.

2) The authors should provide some quantitation on the shift of the position and length of the L2 vein after the enhancer swaps (from other Dipteran species). In particular, the authors should quantify the shift in L2 position seen with the M. domestica CRM (distance of shift after normalizing for variation in wing size) and shortening of L2 seen with the C. capitata CRM (length of L2 relative to total wing length).

We have quantified the *M. domestica* vein displacement phenotype by three different means: 1) measuring the ratio of the length of the L2 versus L3 veins; 2) measuring the angle between the L2 and L3 veins; and 3) measuring the distance (counted in number of thick marginal bristles) distal to the point where the L2 vein intersects the wing margin. All three of these measures reveal highly significant anterior shifts of the L2 vein in the *M. domestica* replacement. With regard to the *C. capitata* replacement, we have measured the length of the rescued L2 vein segment normalized by the length the intact L3 vein. We also quantitated the anterior displacement of the rescued L2 segment relative to that of the wild-type L2 vein in a fixed proximo-distal position.

3) The authors should provide some data on the expression of kni and sal in the wing disks of M. domestica and C. capitata to assess if indeed the pattern of these genes suggests general similarities in the mechanisms of vein placement, and to assess if the pattern seen when the respective CRMs are placed in D. melanogaster is reflective of the normal pattern of expression for kni in the source species.

We have included as supplemental material a figure (Figure 6—figure supplement 1) showing expression of *kni* and *salm* in *C. capitata* wing imaginal disc. These results confirm that these two genes are expressed in a similar pattern in *C. capitata* and *D. mel.* In consultation with the editors, it was decided, in order to avoid a significant potential publication delay, that it would not be necessary to provide comparable data from *M. domestica* given that: 1) the *M. dom.* replacement expresses *kni* in a stripe abutting the *salm* domain in *D. mel.*; and 2) because *kni* is expressed in a similar pattern relative to *spalt* in a yet more diverged fly species (*Megaselia abdita*) as well as in *C. capitata*.

[Editors’ note: after resubmission, the authors sent the note below. The paper was then accepted for publication.]

Since submission of our revised manuscript on November 29, 2017, we have done a stain for Knirps protein expression in *M. domestica* that worked very well, revealing a pattern strikingly similar to what we observed in the *D. mel. riCC-Mdom* CRM replacement. We provide this result in comparison to the *D.mel. wt* and *riCC-Mdom* CRM to help readers appreciate that the broadened stripe we observe in the *riCC-Mdom* CRM replacement most likely reflects the endogenous Kni expression pattern in its species of origin. We have included this stain in the revised version of Figure 6—figure supplement 1.

With these newly added data, we have now addressed all of the reviewer requests.